# Optical Properties of Bulk Single-Crystal Diamonds at 80–1200 K by Vibrational Spectroscopic Methods

**DOI:** 10.3390/ma14237435

**Published:** 2021-12-03

**Authors:** Zitao Shi, Qilong Yuan, Yuezhong Wang, Kazuhito Nishimura, Guojian Yang, Bingxue Zhang, Nan Jiang, He Li

**Affiliations:** 1Key Laboratory of Marine Materials and Related Technologies, Zhejiang Key Laboratory of Marine Materials and Protective Technologies, Ningbo Institute of Materials Technology and Engineering (NIMTE), Chinese Academy of Sciences, Ningbo 315201, China; shizitao@nimte.ac.cn (Z.S.); yuanqilong@nimte.ac.cn (Q.Y.); yangguojian@nimte.ac.cn (G.Y.); 2Center of Materials Science and Optoelectronics Engineering, University of Chinese Academy of Sciences, Beijing 100049, China; 3Advanced Nano-Processing Engineering Laboratory, Mechanical Systems Engineering, Kogakuin University, Tokyo 192-0015, Japan; nishimura@cc.kogakuin.ac.jp; 4Public Technology Center, Ningbo Institute of Materials Technology and Engineering (NIMTE), Chinese Academy of Sciences, Ningbo 315201, China; zhangbingxue@nimte.ac.cn; 5Ningbo New Material Testing and Evaluation Center Co., Ltd., Ningbo 315048, China

**Keywords:** bulk single-crystal diamond, temperature, spectroscopy, IR window

## Abstract

Bulk diamonds show great potential for optical applications such as for use in infrared (IR) windows and temperature sensors. The development of optical-grade bulk diamond synthesis techniques has facilitated its extreme applications. Here, two kinds of bulk single-crystal diamonds, a high-pressure and high-temperature (HPHT) diamond and a chemical vapor deposition (CVD) diamond, were evaluated by Raman spectroscopy and Fourier Transform Infra-Red (FTIR) spectroscopy at a range of temperatures from 80 to 1200 K. The results showed that there was no obvious difference between the HPHT diamond and the CVD diamond in terms of XRD and Raman spectroscopy at 300–1200 K. The measured nitrogen content was ~270 and ~0.89 ppm for the HPHT diamond and the CVD diamond, respectively. The moderate nitrogen impurities did not significantly affect the temperature dependence of Raman spectra for temperature-sensing applications. However, the nitrogen impurities greatly influence FTIR spectroscopy and optical transmittance. The CVD diamond showed higher transmittance, up to 71% with only a ~6% drop at temperatures as high as 873 K. This study shows that CVD bulk diamonds can be used for IR windows under harsh environments.

## 1. Introduction

Diamond is an allotrope of carbon which has attracted much attention because of its excellent physical and chemical properties [1]. Its main applications include optics [2], electron devices [3] and quantum sensors [4] under harsh conditions, such as irradiation, high temperatures, high power and high pressure. These extreme applications of diamonds benefit from their unique properties, which are closely related to their lattice structure and corresponding phonon spectrum [5].

Raman spectroscopy is an effective tool for measuring the crystalline quality of diamonds [6]. Moreover, Raman scattering spectra can reveal a great deal of information about the optical modes of lattice vibration at the Brillouin zone center [7]. There is a triply degenerate LTO optical phonon in the Brillouin zone center corresponding to the sharp first-order Raman line near 1332 cm^−1^ at room temperature. Interestingly, the first-order Raman line regularly shifts as the temperature changes. Investigations of the temperature-dependent Raman spectra in diamonds have been conducted since 1940s, both through experiments [8,9] and theoretically [10,11]. The temperature dependence of the Raman spectra can be attributed to the anharmonic terms in the vibrational potential energy [7,12]. Based on the temperature dependence relationship, Cui et al. [13] accurately measured the temperature of diamond. In fact, as a noncontact temperature measurement method, the temperature dependence of diamond Raman spectra shows has a variety of applications; for example, monitoring the temperature during crystal growth [14] and detecting the surface temperature for studying the interaction of matter [15]. Furthermore, diamond is an ideal choice for infra-red (IR) windows that can survive or withstand extremely harsh environments. Traditional IR materials (such as Ge, ZnS, and ZnSe) show excellent optical transmittance, but the application of these materials is very limited due to low thermal conductivity, poor resistance towards rain and sand erosion [16,17]. The combination of the superior optical, thermal and mechanical properties of diamond makes it an important IR window material [18]. Diamond crystals are also applied for attenuated total reflection (ATR) elements to extend FTIR measurements in harsh environments [19].

Publications on the relationship between Raman spectra and temperature have been widely reported; however, most of these studies mainly focused on natural single-crystal diamonds [8,9]. Additionally, most of the studies on the optical transmittance of diamonds used CVD polycrystalline diamond films or natural single-crystal diamonds [20,21]. The application of natural diamonds is very limited due to high costs. Moreover, most of natural diamonds contain a large number of residual impurities and defects [22]. Recently, great advances have been made in synthesizing bulk diamonds of high quality through the high-pressure and high-temperature (HPHT) method and the chemical vapor deposition (CVD) method [23,24]. The development of synthesizing technology reduces cost and greatly promotes the application of optical-grade diamonds. Therefore, it is necessary to investigate the optical performance of synthesized diamonds—through, for example, Raman spectroscopy and IR transmittance—as well as their temperature dependence relationship. However, few studies have been reported on synthesized bulk diamonds.

In this work, two kinds of bulk single-crystal diamonds, a high-pressure and high-temperature (HPHT) diamond and a chemical vapor deposition (CVD) diamond, were evaluated by X-ray diffraction (XRD), Raman spectroscopy and Fourier Transform Infra-Red (FTIR) spectroscopy at a range of temperatures from 80 to 1200 K. The properties are carefully investigated and compared.

## 2. Materials and Methods

The HPHT single-crystal diamond (3.5 × 3.5 × 1.0 mm^3^) was ordered from Yike Super-hard Materials Co., Ltd. (Shenzhen, China), and the synthesis conditions were ~6 GPa and ~1400 °C. The CVD single-crystal diamond (8.0 × 8.0 × 1.0 mm^3^) was grown by microwave plasma CVD equipment (HMPS-2060SP, HUERAY, Chengdu, China). The substrate was a mirror-polished CVD single-crystal diamond seed of the volume 9.0 × 9.0 × 0.5 mm^3^ (Hubei Carbon Six Technology Co., Ltd., Yichang, China). During CVD diamond growth, the microwave power was 4 kW, and the working pressure was 1.23 KPa. The reaction gas flow rate was 400 sccm, the CH_4_/H_2_ volume ratio was 3%, and the N_2_/H_2_ volume ratio was 0.02%. The substrate temperature was ~1000 °C. Two sides of each sample were carefully polished to attain an optical-grade surface. The surface roughness is <20 nm, evaluated by a laser microscope (VK-X1000, KEYENCE, Osaka, Japan). The samples are shown in Figure 1a,b. The two samples were named HPHT-D and CVD-D for short, respectively. The yellow color of the HPHT-D sample was typically caused by the absorption of light with energy larger than the threshold energy of 2.2 eV (564 nm) [25]. The CVD-D was colorless and highly transparent.

The quality of crystals was characterized by an X-ray diffractometer (D8 ADVANCE, Bruker, Germany) using Cu Kα radiation (1.5406 Å) and a laser confocal Raman spectrometer (LabRAM Odyssey, HORIBA, France) with a resolution of 0.55 cm^−^^1^. Samples were subjected to XRD analysis in the continuous mode with a step size of 0.02° and a step time of 0.1 s over an angular range of 30–140° (2θ). Raman spectra were obtained in backscattering geometry, where a 532 nm Nd: YAG laser was used to excite the diamond samples. To obtain the anti-Stokes lines, the sets of ultralow-frequency notch filters were assembled to the spectrometer. IR and UV–Visible spectra were acquired on a versatile FTIR spectrometer (MB3000, ABB, Quebec City, QC, Canada) and spectrophotometer (Lambda 1050+, PerkinElmer, Waltham, MA, USA), respectively. To investigate the spectral variations at various temperatures, two heating stages were customized from the manufacturer (HUOZI Instrument, Shanghai, China), as shown in Figure 1c,d. The Raman heating stage (HS1) is equipped with a quartz window situated above the sample chamber, and the vertical IR heating stage (HS2) is equipped with two ZnSe windows located before/after the sample chamber. Platinum-rhodium thermocouple is used to measure temperature. Temperature calibration was performed based on the melting point of silver. To collect the low-temperature (80–300 K) Raman spectra, samples were cooled by liquid nitrogen in the heating and freezing stage (THMS600, LINKAM, Tadworth, UK). The measurements of Raman spectra at 300–1200 K were conducted using HS1 under high-purity nitrogen. The measurement of IR transmittance at 25–600 °C was performed using HS2 in the air. The heating and cooling rates were 20 °C/min, with 60 s holding at each temperature. The Raman spectrum was fitted via the Voigt function. In this paper, the FWHM of the Raman peak refers to the Lorentz width.

## 3. Results and Discussion

### 3.1. X-ray Diffraction and Raman Studies

Figure 2a shows the XRD patterns of the samples measured at 300 K. Both the HPHT-D and CVD-D are (100) oriented single crystals with no obvious difference. According to the Raman spectra in Figure 2b, the FWHM of HPHT-D and CVD-D is 1.91 and 1.78 cm^−1^, respectively. The results demonstrate that both the samples have high crystal quality. The larger FWHM value could be attributed to nitrogen impurities in the HPHT-D sample [26].

### 3.2. Studies of N Impurity in Diamonds

Generally, the N atom is the most usual impurity in natural and synthetic diamonds [27]. Its existence forms and concentration will greatly influence the performance of the diamond. Some methods have been suggested to calculate the content of N impurities in diamond [28]. Absorption spectroscopy provides an easy and efficient approach to quickly evaluate the N content. Figure 3a shows the IR spectra of the two types of diamonds. There is very obvious difference between the spectra of HPHT-D and CVD-D at 1400–500 cm^−1^ range. As reported, N impurity is mainly in the form of single substitutional atoms in HPHT-D. The strong absorption peaks at 1130 and 1344 cm^−1^ are related to the C–N bonds. Relative weak absorption peaks at 848 and 1044 cm^−1^ are mainly caused by C–O bonds or other inclusions [29]. In contrast, there is no obvious absorption in CVD-D sample. For both samples, the common absorption peaks at 4000–1500 cm^−1^ are the intrinsic absorption of diamond caused by lattice vibration [30]. According to the quantitative relation between single-phonon absorption intensity at 1130 cm^−1^ and intrinsic two-phonon absorption intensity at 2000 cm^−1^, N concentration in diamond can be calculated by Equation (1):(1)$$\begin{array}{c} N\left(ppm\right)=\left(\frac{\mu_{1130}}{\mu_{2000}}\right)\times \alpha_{2000}\times Lf \end{array}$$
where *μ*_1130_ and *μ*_2000_ are the absorption intensity at 1130 and 2000 cm^−1^, respectively. *α*_2000_ is the absorption coefficient at 2000 cm^−1^ with a value of 12.3. *Lf* (= 25) is the linear factor [30,31]. There is no N-related absorption peak in the FTIR spectrum of CVD-D because of low N content. Therefore, Equation (1) is not suitable for evaluating the N content of CVD-D. Another approach should be explored. The introduction of N could result in characteristic absorption of diamond, as shown in the UV–Visible spectrum (Figure 3b). For CVD-D, an absorption peak could be seen at 270 nm which is caused by the transition of the electron from N impurity energy level to the valence band. Equation (2) is usually used to evaluate the N content [32,33]:(2)$$\begin{array}{c} N\left(ppm\right)=0.56\times \alpha_{270} \end{array}$$
where *α*_270_ is the absorption coefficient at 270 nm. In comparison, HPHT-D has strong light absorption at 564 nm or a shorter wavelength, but the absorption band at 270 nm is not measurable because of the cut-off wavelength of 318 nm due to high N impurities. The absorption is responsible for the yellow color of HPHT-D sample. In this study, the N concentration of HPHT-D is ~270 ppm, while that of CVD-D is ~0.89 ppm. The lower N content of CVD-D is a typical characterization of optical-grade diamond (N content < 1 ppm) [34].

### 3.3. Raman Studies of Diamonds at Various Temperatures

For a better study of the effect of N impurity, Raman spectra of HPHT-D and CVD-D were also collected at low temperature. Figure 4a,b show the Raman spectra of the two diamond samples at 80 K. It could be seen that there is an obvious distinction between the Raman spectra in HPHT-D and CVD-D at 80 K. The CVD-D shows two very strong peaks concerning the zero-phonon line of the nitrogen-vacancy color center (NV^0^ and NV¯) [35]. The intensity of NV^0^ peak is significantly higher than that of the first-order Raman peak (indicated by the arrow in Figure 4b). As temperature rises, the fluorescence background and Raman peaks relative to NV^0^ and NV¯ both weaken because of the enhancing lattice vibration. Figure 4c,d show how Raman spectra change at elevated temperatures. Due to extremely weak signal at low temperature (<300 K), anti-Stokes lines are introduced above 300 K for comparison. The center position of the anti-Stokes peak and Stokes peak are almost symmetric about the zero-shift line, and the FWHM values are the same. The higher the temperature, the wider the peaks of Stokes and anti-Stokes. This phenomenon could be attributed to the shortened optical phonon lifetime caused by the anharmonic terms in the vibrational potential energy [36]. However, the intensity variations of the two peaks are reversed. With an increase in temperature, the Stokes peak intensity declines while the anti-Stokes peak intensity increases, which was caused by the presence of more high-energy ground state phonons. Notably, the Raman peaks of both diamond samples shift from 1332 to 1309 cm^−1^ while the temperature rises from 300 to 1200 K. At the same temperature, the difference in the Raman shift of both samples is lower than 0.8 cm^−1^, indicating that the temperature-dependent Raman shift of the diamond is due to the temperature influence of its intrinsic lattice. Therefore, it is also feasible to determine the temperature by the position of the intrinsic Raman peak of diamond, and the resolution and accuracy of temperature measurement from Raman shift not only depends on the impurity content of diamond, but also the resolution of Raman instrument.

### 3.4. FWHM of Diamond Raman Spectroscopy

The FWHM of the Raman peak reflects information about the lifetime of the zone center optical phonon [11]. The Raman peak is the convolution of the Lorentzian shape of the actual phonons with the response function of the instrument [9,37]. To accurately extract data, the Voigt function (the mixture of Lorentzian and Gaussian components) is used to fit the spectrum. In this study, the spectral resolution of 0.55 cm^−1^ is used for the Gaussian width of Voigt fitting. The fitted results are shown in Figure 5. Compared with Lorentz and Gauss functions, the Voigt function gives the optimum result, especially for the fitting of the two wings of the peak. The fitted FWHM values are 2.03 cm^−1^ (Lorentz function), 2.13 cm^−1^ (Gauss function) and 1.78 cm^−1^ (Voigt function), respectively. It can be seen that choosing different fitting functions will produce different FWHM. Voigt function is more precise based on the fact that the Raman spectrum is the convolution of different signals.

Multiple factors can change the Raman line shape, such as crystal orientation and gas conditions during the growth process [38,39,40]. Studies have indicated that the introduction of N can increase the FWHM value, and hence, shorten the lifetime of Raman phonon [26,41]. From Figure 6, the FWHM of the HPHT-D and CVD-D samples at 80 K are 1.587 cm^−1^ and HPHT-D 1.801 cm^−1^, respectively. Furthermore, the difference between them is very small and remains near constant at elevated temperature. Several sets of data from references are also displayed in Figure 6. Most of the previous research [37,42,43] focused on natural type IIa diamonds. The sample by Surovtsev et al. [5] was from a synthetic IIa diamond, which show similar result to our CVD-D sample. The FWHM of the Raman peak represents the inverse optical phonon lifetime, its temperature dependence shows the anharmonicity of the lattice vibrations. Anharmonicity causes the zone center optical phonon to decay into other phonons that conserve energy and momentum. The cubic term to second order in potential energy means the three-phonon process which makes an optical phonon decay into a pair of acoustic phonons, while the quartic term implies four-phonon interaction causing the annihilation of an optical phonon into three acoustic phonons [7,10,11,44]. For diamond, both the three- and four-phonon processes contribute to FWHM. Assuming that the decayed phonons have the same frequency, the temperature dependence of FWHM could be expressed as Equation (3) [7]:(3)$$\begin{array}{c} \Gamma \left(T\right)=A\left(2n\left(\frac{\omega }{2},T\right)+1\right)+B\left(3n^{2}\left(\frac{\omega }{3},T\right)+3n\left(\frac{\omega }{3},T\right)+1\right) \end{array}$$
where Γ is the FWHM, *T* is the absolute temperature, ω is the phonon frequency (in units of cm^−1^) of the LTO mode, *n* (*ω*,*T*) = 1/[exp ((hc*ω*)/(k*T*) − 1)] is the temperature-dependent Bose–Einstein phonon distribution function, A and B are the adjustable parameters reflecting the relative contributions of three- and four-phonon processes, respectively. The parameter h, c, k are the Planck constant, light velocity and Boltzmann constant, respectively.

### 3.5. Anti-Stokes and Stokes at Various Temperatures

Raman shift and the intensity ratio of anti-Stokes line (*I_aS_*) to Stokes line (*I_S_*) are also temperature-dependent (see Figure 7a,b). At low temperatures, the phonon frequency is nearly constant. It can be seen that the temperature-dependent law of the zone center optical phonon frequency and *I_aS_*/*I_S_* is very consistent between HPHT-D and CVD-D. This suggests that N impurities have a faint effect on the temperature dependence of Raman shift and *I_aS_*/*I_S_*. Similar to FWHM, the model of Raman shift versus temperature could be given by Equation (4) [7]:(4)$$\begin{array}{c} \Omega \left(T\right)=\omega_{0}+C\left(2n\left(\frac{\omega }{2},T\right)+1\right)+D\left(3n^{2}\left(\frac{\omega }{3},T\right)+3n\left(\frac{\omega }{3},T\right)+1\right) \end{array}$$
where *ω*_0_ is Raman shift at 0 K, *C* and *D* are adjustable parameters. The intensity of the anti-Stokes line will significantly increase with rising temperature. It is comparable to the intensity of the Stokes line at high temperatures. All of them are Bose–Einstein statics related. The relationship between the intensity ratio of the anti-Stokes peak to Stokes peak and temperature is given by the following Equation (5) [13]:(5)$$\begin{array}{c} \frac{I_{aS}}{I_{S}}={\left(\frac{\omega_{l}+\omega }{\omega_{l}-\omega }\right)}^{4}\gamma \frac{1}{exp\left(\frac{hc\omega }{kT}\right)} \end{array}$$
where *ω*_l_ refers to the incident laser frequency and *γ* is the correction factor. The influence from absorption, reflection, and matrix element changes for the two different Raman scattered photons are contained in *γ* [45].

By definition, the relationship between Γ, Ω, *I_aS_/I_S_*, and reduced temperature (k*T*)/(hc*ω*) should be established. Generally, most studies treat the model with ω_0_ as ω due to the small difference. In this work, *ω*_0_ is assigned to 1333 cm^−1^, which is the maximum value of our test results and used as *ω*. According to the fitting curves of CVD-D (Figure 6 and Figure 7), the corresponding parameters from the nonlinear fitting of our data using Equations (3)–(5) are listed in the inserted tables. The temperature dependence of Raman shift can be guided for temperature-sensing applications by establishing accurate models [46]. As is well known, diamond has ultra-thermal conductivity up to ~2000 W/(m·K), ultra-high melting-point ~4000 °C, ultra-low thermal expansion coefficient ~1.0 × 10^−6^/K, and ultra-high hardness ~100 GPa [47]. The numerical fitting from measurement or data processing is very crucial, since a small numerical deviation could cause a large temperature difference.

### 3.6. IR Transmittance Studies of Diamond at Various Temperatures

When a diamond is used for an IR window, one of the most concerned properties is the optical transmission. Thus, we studied the high-temperature IR transmittance of HPHT-D and CVD-D at the 3–12 μm band, as shown in Figure 8. Diamond has strong absorption in the 2.5–6.0 μm region, which was caused by two- and three-phonon absorption [50]. Therefore, diamond is not a good choice for a mid-wavelength IR (3–5 μm) window application. At the long-wavelength IR (8–12 μm) range, HPHT-D has lower transparency because of N-induced one-phonon absorption [30]. In contrast, CVD-D exhibits the theoretical transmittance in the long-wavelength band at 298 K. At higher temperature, both the transmittance of HPHT-D and CVD-D decrease, resulting from enhanced phonon absorption. No new characteristic absorption peak appears. The transmittance of CVD-D at 873 K is reduced by 6% on average within the 8–12 μm region compared to the value (71%) at 298 K. Furthermore, CVD-D show excellent optical property and is more suitable for long-wavelength IR window applications in comparison with ZnS. Since high-temperature transmittance of ZnS sharply drops at >9 μm band [51] as a result of the intensified lattice vibrations. It is therefore suggested that only those with less impurities such as CVD diamonds can be used for IR windows under harsh environments benefiting from the combination of high transparency, and remarkable thermal and mechanical properties.

## 4. Conclusions

In summary, vibrational spectroscopy was used to investigate HPHT diamonds and CVD diamonds over a range of temperatures. The nitrogen content of the two diamonds evaluated by spectroscopic methods was ~270 ppm for the HPHT diamond and ~0.89 ppm for the CVD diamond, respectively. The temperature dependence of the center position, FWHM and the intensity ratio of anti-Stokes to Stokes of Raman peak are compared in detail. The relationships between the studied Raman parameters and temperature can offer effective references for temperature-sensing applications. Numerical fitting from measurement or data processing is crucial, since a small numerical deviation could cause a large temperature difference. Compared with the HPHT sample, the CVD diamond shows higher IR transparency due to less nitrogen impurities. The CVD diamond showed higher transmittance, up to 71% with only a ~6% drop at temperatures as high as 873 K. The present work shows that both HPHT and CVD diamonds are suitable for temperature-sensing applications. CVD bulk diamonds can be used for IR windows under high-temperature and harsh environments. Great advances can be anticipated with the development of optical-grade bulk diamond synthesis techniques in the future.

## Figures and Tables

**Figure 1 materials-14-07435-f001:**
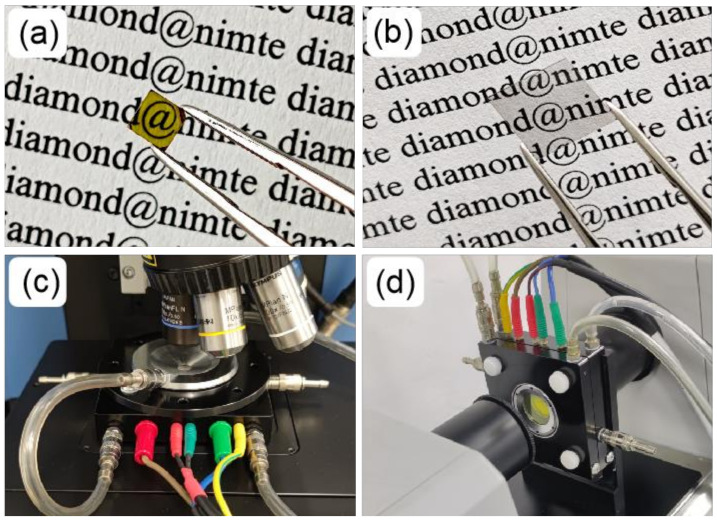
Photographs of (**a**) HPHT-D, (**b**) CVD-D, (**c**) the Raman heating stage and (**d**) the IR heating stage.

**Figure 2 materials-14-07435-f002:**
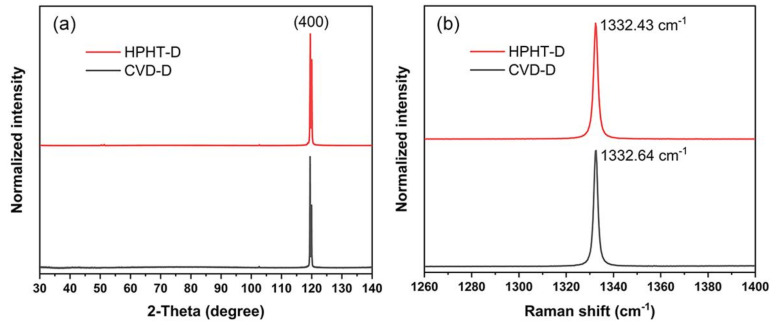
(**a**) XRD patterns, and (**b**) the first-order Raman spectra of HPHT-D, CVD-D at 300 K.

**Figure 3 materials-14-07435-f003:**
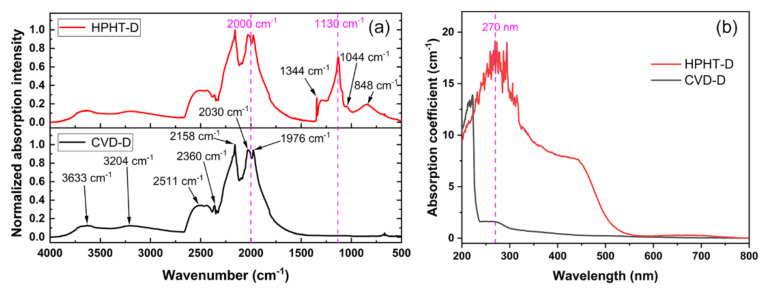
(**a**) FTIR spectra and (**b**) UV–Visible spectra of the HPHT-D and the CVD-D samples.

**Figure 4 materials-14-07435-f004:**
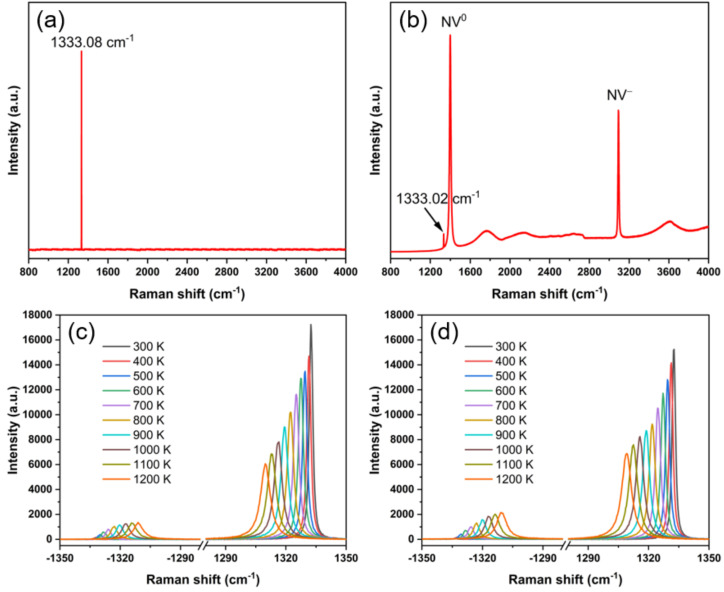
Raman spectra of (**a**) HPHT-D, (**b**) CVD-D at 80 K and the anti-Stokes and Stokes peaks of (**c**) HPHT-D, and (**d**) CVD-D at higher temperature. The first-order Raman peak in (**b**) is pointed out by the arrow.

**Figure 5 materials-14-07435-f005:**
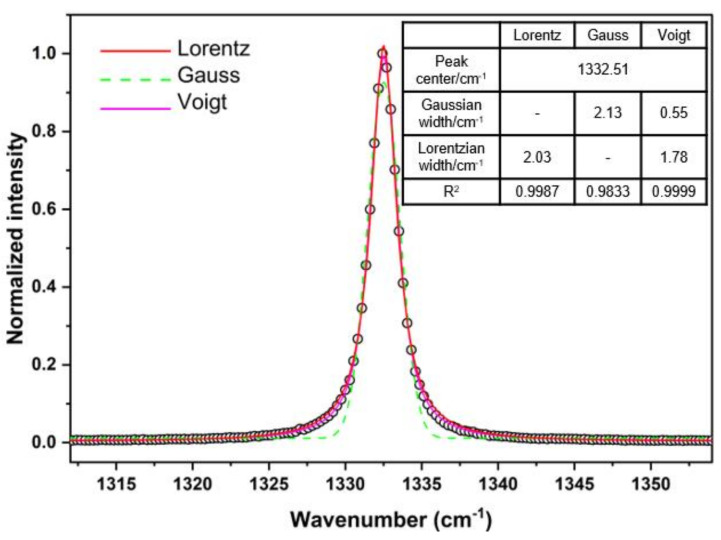
The first-order Raman spectrum of CVD-D at 300 K (circles). The data are fitted by Lorentz, Gauss, and Voigt functions, respectively. The fitting parameters are listed in the inserted table.

**Figure 6 materials-14-07435-f006:**
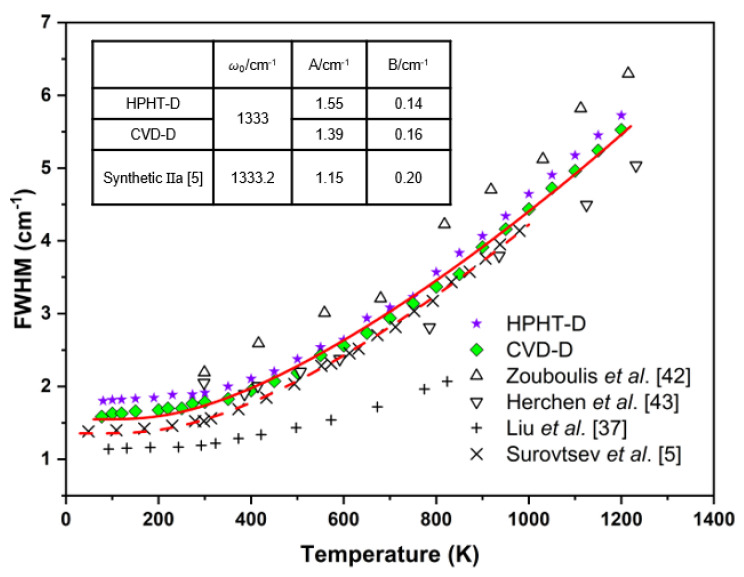
Temperature dependence of the FWHM in diamonds. The solid line is the fitting curve of CVD-D, and the dashed line is the result from Surovtsev et al. [5]. The fitting parameters are listed in the inserted table.

**Figure 7 materials-14-07435-f007:**
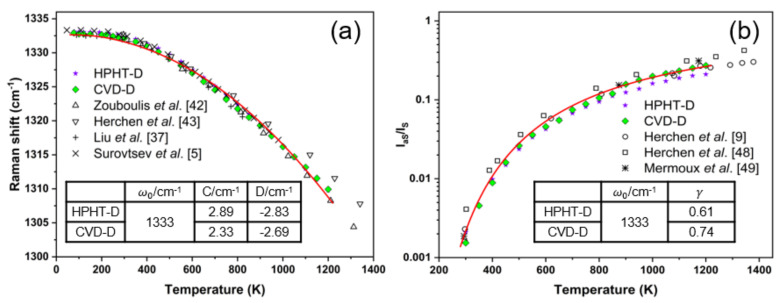
Temperature dependence of (**a**) the frequency for the first-order Raman mode; (**b**) the anti-Stokes to Stokes intensity ratio. The solid line is the fitting curve of CVD-D. The fitting parameters are listed in the inserted tables. Some data from Refs. [5,9,37,42,43,48,49] are also shown for comparison.

**Figure 8 materials-14-07435-f008:**
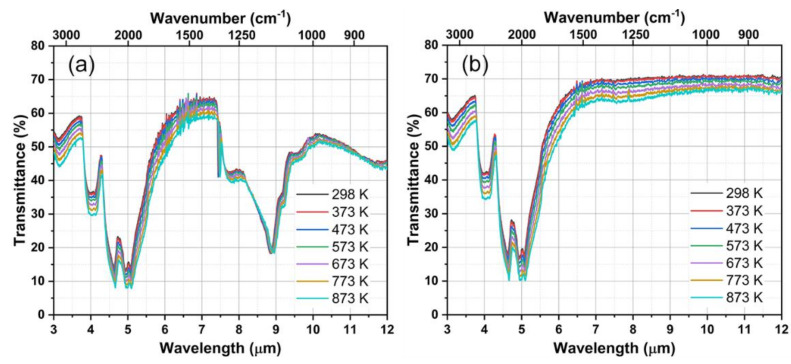
The high-temperature IR transmittance of (**a**) HPHT-D and (**b**) CVD-D measured in the air.

## Data Availability

Data are contained within the article.

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
