# Peer review of "Optical Properties of Bulk Single-Crystal Diamonds at 80–1200 K by Vibrational Spectroscopic Methods"

_materials, 2021, doi:10.3390/ma14237435_

Round 1
Reviewer 1 Report
This is an extremely interesting topic. The authors present their study well; the abstract is up to the point, and the introduction is very useful focusing in the problem. Some points:
- Lines 50-58. Diamond crystals for ATR applications present further advantages, mainly because they can be used further for F-IR measurements.
- Line 74: Please, write XRD in full, I think that this is its first mention.
- Line 87: XRD measurement conditions are missing (step size, time per step, angle range)
- Figure 3a: FTIR spectra are usually depicted from high to low wavenumbers (4000-500 cm-1)
- Figure 8. As the spectra under study concern part of the mid-IR region, consider adding a wavenumber axis.
Please, check minor template and grammar issues, such as presentation of captions. I have highlighted inside the manuscript’s pdf some points.

Reviewer 2 Report
Authors report results on vibrational and optical properties for synthetic diamonds grown by two different methods. Authors did a good work with all the measurements and data presentation. Results are quite clear and are in agreement with ones published before. Authors need to highlight the novelty of their work. Similar experiments on synthetic diamonds with very low nitrogen content were performed before. Authors devote a whole section to measuring nitrogen concentration in their samples, but do not involve this information in the discussion on the main topic of the paper. I do not think that paper can be published in its presents form.
- The title is a bit confusing. Article reports optical absorbtion/transmittance, which are optical properties measured by optical methods, while Raman shift (the main topic of the present paper) is not an optical property in a way most reader would expect it. "Variety of temperatures" sounds clunky. Maybe it is better to write the range "80-1200 K".
- Please provide more information on the samples. You stress that these samples might be interesting to study, so explain them better. For HPHT add the information from the supplier. For CVD explain in more details the growing conditions.
- Can you compare calculated nitrogen content in HPHT obtained from both methods (eq.1 and eq.2)? Are numbers very different?
- In Sec. 3.3, Fig. 4B a peak related to NV0 is shown, while any signs of this defect are absent in optical spectra (Fig. 3B). Can you explain?
- "It can be seen that the selection of fitting functions has a great influence on the FWHM value" Isn't that a function's property, independent from the data?
- The fact that transmittance/FWHM/Raman shift changes with temperature is well-known. It would be interesting and add something to the work, if authors would try to estimate the resolution and accuracy of possible temperature measurements from Raman shift.
- "It is therefore anticipated that, the relationships between the studied Raman parameters and temperature offer effective references for temperature sensing applications."
"It is therefore suggested that CVD diamond can be used for IR window under harsh environments benefited from the combination of high transparency, remarkable thermal and mechanical properties."
These statements are well-known facts about diamond. - Discussion and conclusions do not provide any new insights on the problem, except that it was measured on the new type of samples.
Reviewer 3 Report
The work presents optical properties of 2 types of diamonds at a variety of temperatures.The article shows FTIR, Raman and XRD measurment under a variety of temperatures from 80 K to 1200 K.
The article is written very well. I only have a couple of questions.
The authors present a very strong peak from NV0 for CVD in the Raman spectrum. Was there a visible peak from NV- about 3130 cm-1?
Did the peak from NV0 change with temperature (amplitude)?
References - Double numbering e.g. line 313
Round 2
Reviewer 2 Report
Point 3. It is very confusing to have two spectra on one figure (3b), that were measured with different setups, so they are not comparable. You should explain why did you use different setup and add it to the caption. Why measure 270nm line for N concentration calculations with a setup that cannot measure 270 nm line?
So nitrogen content for each sample is calculated by different methods? This should be mentioned in the text, it is not clear now. Now when you added that peak shift is connected to the N concentration it would be great to have it calculated precisely with both methods.
Point 4. Stimulated emission and optical absorption are two very different things. NV0/NV- defects have known optical absorption lines around ~2 eV. And the defect is always there: at 80 K and 300 K.
Point 6. So peak shifts for 23 cm-1 for 900 degrees, roughly 0.26cm-1 per 10 degrees. For very good Raman spectrometer with resolution 0.8 cm-1 accuracy would be far below average IR thermometer...
